# Operational Stress Injury

**Abraham Rudnick** [1,2,*] **, Andrea Shaheen** [1] **, Sarah Lefurgey** [1] **and Dougal Nolan** [1]

[1] Nova Scotia Operational Stress Injury Clinic, Nova Scotia Health Authority, Dartmouth, NS B3B 1Y6, Canada; andrea.shaheen@nshealth.ca (A.S.); sarahlefurgey@gmail.com (S.L.); dougal.nolan@nshealth.ca (D.N.)

[2] Departments of Psychiatry and Bioethics and School of Occupational Therapy, Dalhousie University, Dartmouth, NS B3B 1Y6, Canada

[*] Correspondence: abraham.rudnick@nshealth.ca

**Definition:** An operational stress injury (OSI) is a term used most often to describe mental disorders which result from, or are exacerbated by, military or police service. In the Canadian context, this most often refers to active or former members of the Canadian Armed Forces (CAF) or the Royal Canadian Mounted Police (RCMP). The most common diagnoses within this term include post-traumatic stress disorder (PTSD), anxiety disorders, depression, and substance use disorders.

**Keywords:** mental health; operational stress injury; post-traumatic; recovery



## 1. Introduction

Operational stress injury (OSI) is a non-clinical and non-medical term apparently formalized by Veterans Affairs Canada [1] yet applicable elsewhere and used to describe a distinct category of occupational stress injury, referring to mental health challenges related to military or police service (the latter reserved in Canada for service in the Royal Canadian Mounted Police (RCMP)) [2]. Occupational stress injuries encompass a broad range of psychological and related conditions resulting from work-related activities that interfere with a person's professional and personal life, including anxiety, depressive, trauma- and stressor-related, and substance-related disorders [3,4]. Operational stressors that may result in OSI can include service-related events beyond combat, such as service-related accidents in unrelated to combat activities, as well as sexual trauma caused by peers and others involved in one's service. OSIs do not relate to injuries that are not associated with service. For example, grief from the loss of a loved one outside the service during active service is not typically a cause of OSI, although if it is associated with lack of support by service peers and/or supervisors, that can cause OSI (sometimes termed consequential OSI). Due to the common stress hypersensitivity related to post-traumatic and other OSI-related experiences, people with OSIs often develop anxiety and substance-related disorders, e.g., as learned stress intolerance and as (maladaptive) self-help coping, respectively.

The recognition of operational stress injuries (OSIs) traces back to earlier periods in history, with notable observations of the psychological impact of combat experiences on soldiers dating back to the mid-19th century. During World War I, the notion of shell shock was addressed, highlighting the psychological toll of warfare on military personnel [5]. However, it was later in the 20th century when the term "post-traumatic stress disorder" (PTSD) was coined, reflecting a deeper understanding of the condition and its broader manifestations beyond the context of combat experiences [6]. Key breakthroughs in the late 1990s and early 2000s furthered awareness of mental health issues in the military and law enforcement communities. Research efforts shed light on the prevalence and impact of OSIs, paving the way for proactive support approaches and comprehensive programs focused on OSI prevention, diagnosis, and treatment. While the term OSI is most commonly used and understood in Canadian settings when referring to military or police service, it is also relevant internationally and to other first responder populations. In any cultural context, as

long as the injury develops during, as a result of, or is worsened by, military/police service, then it can be considered an OSI.

Despite progress, fully comprehending OSI's scope and complexity remains a challenge and hence disrupts related care. OSIs encompass various mental disorders as noted above [1]. The lack of a coherent and comprehensive description of OSI in the peer-reviewed literature requires further scholarly work and publication. OSIs are common among military personnel and RCMP members, often co-occurring with physical health challenges [7]. The study of OSIs is important due to their considerably disruptive impact on individuals and their social supports [8], including considerable suffering and dysfunction; as well, it is important to address related resilience and post-traumatic growth [9].

OSI remains a critical issue requiring ongoing research and collaboration to effectively support military personnel, veterans, and RCMP (and other police forces') members and retirees. Understanding OSI's history and current status is important for further research and development including in relation to best practices in order to reduce incidence and enhance recovery in relation to OSIs.

This entry aims to provide a comprehensive yet concise overview of OSIs, encompassing diagnosis-specific and transdiagnostic aspects. By examining symptomatology, epidemiology, etiology, treatment, and lived experience, valuable insights will be offered here for all stakeholders, such as mental health care providers, policymakers, and service users.

## 2. Diagnoses and Symptoms

OSIs most commonly include trauma and stressor-related disorders, anxiety disorders, depressive disorders, and substance use disorders [10]. In addition, mental and physical comorbidities are common as part of the health challenges of people with OSIs. This section describes common mental disorders that comprise OSI, as well as their mental and physical comorbidities, using current diagnostic classification from the DSM-5TR and/or ICD-11 [11,12] and disorder-specific as well as transdiagnostic descriptions of symptoms and related matters such as impact on the person's functioning.

### 2.1. Trauma- and Stressor-Related Disorders

Trauma- and stressor-related disorders are perhaps the most common and well-known type of mental disorder that can be an OSI. In this category, the three mental disorders that appear to be most commonly addressed are PTSD, acute stress disorder (ASD), and other specified trauma- and stressor-related disorder (OSTSRD). In general, individuals in military, police, or other first responder jobs can be exposed to major, frequent, or prolonged stressful events throughout the course of their employment. We know from animal-based research that all of these types of events can elicit stress and or/PTSD-like responses, which likely explains the prevalence of this disorder category within the OSI population [13]. PTSD involves one or more traumatic events (traumas) that are experienced, witnessed, or learned about by the person exposed to them [11]. Such a trauma typically involves harm or risk to life, limb, organ, or other key parts of one's well-being; an OSI often involves traumas such as intentional and unintentional killing and physical injury, suicide and other severe self harm, and physical and sexual abuse. A couple of caveats are that witnessing a trauma typically has to occur in a person's own life (rather than removed from their own life such as in a movie) and learning about a trauma has to be about someone known to the person (typically a significant other). Some evidence suggests that there may be lower levels of secondary or learned trauma amongst first responder populations compared to the general population, but this could be confounded due to difficulties in sampling and fears of job loss amongst these populations [14]. Symptoms of PTSD have to include intrusive symptoms that refer to re-experiencing the trauma (such as memories of it, related nightmares, flashbacks that consist of re-living the trauma, and dissociation from reality in the moment such as out of body experiences, sometimes triggered by stimuli such as sights and sounds that remind the person of the trauma); avoidance of such triggering

stimuli (internal/cognitive avoidance, such as distraction, and external/social avoidance, such as of crowds and other stressful situations); negative thinking and low mood (such as mistrust of people, hopelessness, lack of energy and motivation including to help oneself, and suicidal thinking); and hyperarousal and hyperreactivity to regular and other events (such as being irritable, easily startled, having poor concentration and at least partly in relation to that having impaired short term memory, and experiencing insomnia—especially waking up repeatedly at night). These symptoms have to last for at least a month to be diagnosable as PTSD, otherwise they would be considered ASD (which is similar to PTSD in its symptoms although ASD more commonly manifests with dissociative symptoms) [15]. OSTSRD typically involves some but not all of these symptoms or all of them but no clear trauma (rather the related stressful experience may be of lesser severity but still disruptive, such as experiencing work-related harassment); not much is known yet about OSTSRD [16].

### 2.2. Depressive Disorders

Another type of mental disorder that can be considered an OSI refers to depressive disorders. In this category, the two mental disorders that appear to be the most commonly addressed are major depressive disorder (MDD) and persistent depressive disorder (PDD). MDD involves one or more major depressive episodes (MDEs), which refers to a set of symptoms that are often but not always time limited. An MDE consists of some or all of the following symptoms (all or most of the time): depressed mood (often with hopelessness), reduced interest or pleasure in regular activities, reduced appetite and/or weight, insomnia (or the opposite—hypersomnia, which is much sleep), movement decrease or increase, fatigue or loss of energy, feeling worthless or unusual guilt, poor concentration or indecisiveness, and recurrent thoughts of death and/or self-harm. PDD presents with similar but fewer symptoms that last at least two years, unlike MDD, which is typically shorter lived (MDD's MDEs have to last at least two weeks).

### 2.3. Anxiety Disorders

Anxiety disorders can also be considered an OSI. In this category, the two mental disorders that appear to be the most commonly addressed are panic disorder (PD) and generalized anxiety disorder (GAD). PD involves one or more panic attacks, each panic attack typically lasting a few minutes to a very few hours (with some of the symptoms such as fast heart rate, sweating, shaking, shortness of breath, chest pain, nausea, dizziness, feeling hot or cold, and fear of loss of control and of death); in addition, the person is concerned about experiencing one more panic attack in the future and/or exhibits related maladaptive behavior, such as avoidance of situations that may trigger panic attacks (such as crowds; notably, if a person with an OSI experiences one or more panic attacks but is not concerned about one or more future panic attacks nor related maladaptive behavior, they have panic attacks without panic disorder, which can occur as part of PTSD). GAD involves difficult to control worries (for most of the days of at least six months) in relation to at least two types of events or activities (such as health, money, school, and work), with some of the following symptoms: restlessness, fatigue, poor concentration, irritability, muscle tension, and insomnia or restless sleep.

### 2.4. Substance-Related Disorders

Substance-related disorders are often an OSI that encompass both substance use disorders and substance-induced disorders. Substance use disorders are characterized by intense craving for the substance and sometimes engaging in risky behaviors to obtain or use the substance. On the other hand, substance-induced disorders refer to the mental health complications, such as anxiety, depressive, neurocognitive, and psychotic disorders, that may arise as a result of substance use. In relation to this category, the prevalence of its disorders varies depending on the substances most popular in different jurisdictions; some of the most common substance-related disorders across much of the world are alcohol-related disorder, cannabis-related disorder, opioid-related disorder and tobacco-

related disorder (other substance-related disorders such as sedative-related disorder and stimulant-related disorder are also fairly common; notably, polysubstance-related disorder—referring to misusing more than one substance—is common too). Although substance-related disorders can develop as part of maladaptive coping with other mental disorders, if they persist, they pose distinct challenges that require addressing separately. Optimally, if they are thus comorbid, treatment for them and the other mental disorder(s) a person is experiencing would be provided concurrently.

All of these mental disorders involve distress and/or dysfunction of the person with the OSI (such as disrupted activities of daily living, work challenges, and ineffective in any social interactions). A key aspect of human life, i.e., family relations, is often particularly disrupted by an OSI, such as marital and other family discord as well as parenting challenges; with that being said, families who continue to support people with OSIs are not unusual. Other social supports such as friendship are also often disrupted, which is compounded by the fact that some military or police members or veterans (all collectively termed "peers") of the person are themselves experiencing an OSI. This is also compounded by the fact that interacting with such peers can remind the person with the OSI of their military or police work-related traumatic experiences and hence trigger post-traumatic symptoms of the person with an OSI which often leads that person to avoid such peers.

*2.5. Physical Comorbidities*

Comorbidities related to OSIs can be mental or physical. In addition to the above noted mental disorders that can present as an OSI, as well as its comorbidity (such as when substance use disorders develop as a maladaptive way of coping with the distress related to an OSI), some physical disorders are particularly common [17–20] as comorbidities of OSIs. Musculoskeletal disorders often co-occur with OSIs due to causes such as physical injuries that sometimes co-occur with mental traumas (such as in combat and other assaults as well as in motor vehicle accidents) and muscle tension that accompanies post-traumatic stress symptoms. Cardiovascular disorders also often co-occur with OSIs due to causes such as the reduced range of heart rate variation in relation to regularly fluctuating stress that may lead to hypertension and ischemic heart disease. In addition, dental disorders also often co-occur with OSIs due to causes such as bruxism (teeth grinding and jaw clenching) that may lead to complications such as cracked teeth and persistent headaches.

**3. Epidemiology and Etiology**

The prevalence of protective and risk factors—rather than causes, considering that causes of mental disorders are not well-known—that are related to the development and to the persistence of OSIs vary considerably, partly in relation to the various mental disorders that compose OSIs. Yet there are pertinent common factors associated with all OSIs. In particular, women and some special populations such as people of color may experience higher rates of prevalence and risk factors of OSIs compared to the general population, apparently at least partly related to experiencing social discrimination and related behaviors such as sexism and racism [21–23]. Notably, there is much overlap of risk factors for the development of OSIs and of risk factors for the persistence of OSIs (and of its biopsychosocial complications and other consequences). In this section, we address PTSD, as it is arguably the most typical OSI, although other types of OSIs are still more prevalent in the population of armed forces veterans than in the general population [24].

The lifetime prevalence of PTSD ranges considerably with an average of nearly 10% of the general population in developed countries such as Canada; it depends largely although not exclusively on social factors [25]. Military and police members and veterans who have experienced traumas in hostile and/or remote military or police settings are more likely to develop PTSD [26,27]. More generally, some pre-trauma factors, trauma-associated factors, and post-trauma factors are associated with onset and outcome(s) of PTSD (particularly PTSD that is an OSI). Some related pre-trauma risk factors are developmental (childhood)

trauma, pre-existing mental disorders, and physiological (such as food and sleep) deprivation soon before the index trauma; the latter is particularly common in relation to combat warfare. Some related trauma-associated risk factors are experiencing intentional (rather than accidental) trauma that is perpetrated by humans, and co-occurring head injury (even if due to the head injury the person does not remember much if any of the traumatic event) [28]. Some related post-trauma risk factors have comorbidity and poor social support [29]. These factors address higher risk. It is not well-known what protective factors delay or prevent the onset and outcome(s) of such PTSDs (although apparently many if not most people exposed to such traumas do not develop PTSD, as is evident in relation to combat warfare); yet repeated exposure to trauma such as combat warfare increases the risk of developing PTSD [30]. The outcome of PTSD is such that many people with it continue to experience some or many of its symptoms, which is particularly evident in relation to people with PTSD that is an OSI, as, depending on the type of treatment provided, between 38-60% of them show significant improvements following treatment [31–33].

## 4. Biopsychosocial Care

OSIs present significant challenges to the mental health and well-being of people experiencing them. As noted above, OSIs encompass a range of mental disorders, particularly trauma- and stressor-related disorders, anxiety disorders, depressive disorders, and substance use disorders, which frequently co-occur with physical health challenges [1,2]. Given the complexity of OSIs, biopsychosocial care that is evidence-based/informed and person-centered [34] is needed for people experiencing OSIs. Core interventions used for OSIs include evidence-based psychotherapies and psychotropic medications to address diverse symptoms associated with these mental disorders. Other evidence-based/informed interventions such as neuromodulation and psychiatric psychosocial rehabilitation (including cognitive remediation) sometimes help further address OSIs and their correlates.

### 4.1. Psychotherapies

Evidence-based psychotherapies alleviate or reduce symptoms and their disruptive impacts and thus promote clinical recovery. Among the diverse range of OSIs experienced by Canadian veterans, PTSD is typically the most common mental disorder [20]. While there are various treatments for PTSD, prolonged exposure (PE), cognitive processing therapy (CPT), and trauma-focused cognitive behavioral therapy (TF-CBT) have been strongly recommended as first line treatments for PTSD in recent treatment guidelines [35].

Exposure-based therapies, such as prolonged exposure (PE), are deeply rooted in emotional processing theory, as proposed by Foa and Kozak in 1985 and 1986. This suggests that traumatic experiences create problematic fear structures in memory, leading to incorrect associations, exaggerated fear responses, and misunderstandings about safety. PE involves confronting traumatic memories to activate these fear structures and then incorporating new, healthier information to correct these associations. This process helps reprocess traumatic memories and alleviate PTSD symptoms, making exposure-based therapies vital for trauma recovery [36]. PE, CPT, and TF-CBT have a large evidence base and are trauma-focused, which means they directly address memories of the traumatic event or thoughts and feelings related to the traumatic event. Exposure-based therapies have the largest and strongest research evidence base [37], and research and meta-analyses comparing PE, CPT, and trauma-focused CBT do not find that one treatment outperforms the other [37,38].

An intrusive symptom frequently observed among individuals diagnosed with PTSD is the manifestation of trauma-related nightmares (TRN). While there may be some reduction in TRN as a result of CPT or PE, sleep-specific interventions are often needed [39] and some medications can be helpful (which are covered in the next section of this document). The American Academy of Sleep Medicine initially endorsed image rehearsal therapy (IRT) as a recommended treatment for TRN [40]. However, a subsequent update in 2020 introduced cognitive behavioral therapy for insomnia (CBT-I) as an alternative therapeutic

approach for addressing this symptomatology [41]. Despite these advancements, a degree of uncertainty persists regarding the most efficacious interventions for TRN, highlighting the exigency for further comprehensive investigation and research in this particular area.

Furthermore, CBT has been recognized as a widely effective treatment for other mental disorders that are OSIs, such as mood, anxiety, and substance use disorders. For mood disorders, such as MDD and PDD, CBT and interpersonal therapy (IPT) are effective [42]. Exposure-based therapies, such as exposure therapy, are effective in treating anxiety disorders, including GAD and PD [43]. In addition, cognitive-behavioral interventions as well as motivational interviewing (MI) effectively address substance use disorders [44].

Mindfulness-based interventions, such as mindfulness-based stress reduction (MBSR) and mindfulness-based cognitive therapy (MBCT), have shown promise in reducing symptoms of anxiety and depression, as well as enhancing overall well-being [45].

Norris et al. conducted research that suggested causality between PTSD in the veteran and adverse family effects [46]. PTSD was found to have negative effects on family functioning and overall well-being of the family members, including but not limited to relationship distress and lower family cohesion [47]. Despite these findings, a scoping review found that family support yields positive outcomes for veteran mental health and well-being [48]. Emotion-focused therapy (EFT) for couples has been shown to be effective for mitigating relationship distress in veteran couples. Ganz et al. explored EFT's impact on veterans and their partners, addressing PTSD, depression, and relationship distress [49]. Positive changes emerged, boosting relationship satisfaction and reducing depression symptoms post-intervention. Notably, even participants meeting clinical criteria for PTSD, depression, and relationship distress benefitted. EFT showcases its value for enhancing relationships and individual well-being for veterans and their spouses.

Cognitive behavioral conjoint therapy (CBCT) is an additional intervention which may improve PTSD symptoms and the functioning of intimate relationships byway of treating an individual with PTSD while including their intimate partner throughout the intervention [50]. This intervention has demonstrated some efficacy with relation to improving PTSD symptoms in relationship functioning, but more research is needed to identify its effectiveness [51].

### 4.2. Medications and Neuromodulation

Psychotropic medications are beneficial for some of the symptoms of OSIs. For example, antidepressant medications such as selective serotonin reuptake inhibitors (SSRIs) and serotonin and norepinephrine reuptake inhibitors (SNRIs) as well as older antidepressant medications such as tricyclic antidepressants (TCAs) and monoamine oxidase inhibitors (MAOIs) often reduce the severity of post-traumatic stress, anxiety, and depressive symptoms [35,52–54]. Some of these medications assist with some of the comorbidities of OSIs, for example, some SNRIs and TCAs sometimes alleviate or reduce the severity of neuropathic physical pain [55]. Other psychotropic medications such as alpha 1 antagonists such as prazosin reduce the severity of, or eliminate, trauma-related nightmares [56,57]. For persistent insomnia, other psychotropic medications such as quetiapine (at low doses) or trazodone assist. Anxiolytics and sleep inducers such as benzodiazepines and z drugs are best avoided if possible, at least for the long term due to their adverse effects (including gradual development of tolerance for them) [58]. In addition, for some substance-related disorders, particularly those associated with use of alcohol and/or opioids, medications such as naltrexone are sometimes helpful to achieve and maintain abstinence [59,60]. Risk of medications has to be addressed, such as in relation to adverse effects and complications, including both common adverse effects such as sexual dysfunction and uncommon complications such as serotonin syndrome. Notably, dissociative symptoms that are commonly part of PTSD do not respond to medications.

Recently there is a recurrence of trials of psychedelic and related treatments for OSIs. Ketamine is effective for persistent depression [61]. Psylocibin and other psychedelics are showing promise for depression and/or PTSD, particularly when combined with psy-

chotherapy [62,63]. Neuromodulation uses electric or magnetic stimulation to change brain functioning. In relation to OSIs, repetitive transcranial magnetic stimulation (rTMS) often improves persistent depression [64]. Neuromodulation for PTSD is still fairly experimental.

*4.3. Additional Interventions*

In addition to evidence-based psychotherapies and psychotropic medications, other interventions have shown promise as adjunct treatments in treating OSIs and enhancing overall well-being. Social support has an important role in the recovery of people with OSIs. As part of that, peer support provides understanding, empathy, and a sense of belonging, which are important for adaptive coping with the aftermath of traumatic experiences [65]. Military veterans have identified benefits from peer support, include gaining more purpose and meaning in life, normalization of symptoms, enhanced hope, and more [66].

Cognitive remediation (CR) has emerged as a valuable set of interventions for people with mental disorders, addressing (neuro)cognitive impairments that are associated with but distinct from mental symptoms; CR also addressed (neuro)cognitive impairments resulting from traumatic brain injuries (TBIs) that are often comorbid with OSIs. CR aims to improve (neuro)cognitive abilities such as attention, memory, and problem-solving, which are often—although not always—disrupted for people with OSIs [67]. By addressing such (neuro)cognitive impairments with CR, people with OSIs often experience enhanced cognitive flexibility and more adaptive coping strategies, enhancing their overall functioning and quality of life. Notably, CR is more effective when combined with other PSR such as supported employment; PSR in general addresses (residential, social, vocational, and other) environments of the person's choice and primarily involves enhancement and maintenance of adaptive skills and supports [68].

Yoga and other complementary and alternative medicine (CAM) interventions have gained popularity as adjunct treatments added to core treatments for OSIs. Such CAM practices emphasize self-awareness, stress reduction, and emotion regulation. Yoga, with its focus on breath control, meditation, and physical postures, has been associated with reduced anxiety, improved mood, and enhanced resilience [69]. These mind–body practices provide people with OSIs valuable tools to manage their stress and enhance their well-being.

Creative arts therapies, including art therapy, music therapy, and dance/movement therapy, offer distinct approaches for individuals to process traumas and express emotions. Art therapy has become increasingly acknowledged as a complementary intervention for military veterans with PTSD and TBIs [70].

There have also been a variety of programs designed to help veterans with mental disorders re-integrate into the workforce after military service as it is known that work functioning can be impaired in this population [71]. While some of these programs have shown positive results with respect to improved work functioning [72,73] there have also been reported challenges with enrollment [74] and long-term employment outcomes [75]. These programs may be helpful for some of the population with OSIs; however, further research is needed to ensure they are efficacious and produce positive outcomes, e.g., use of work-related accommodations for people with mental health challenges such as time flexibility and social support at the workplace has been shown to be effective but has addressed people with severe and persistent mental illness such as schizophrenia more than people with OSI [76].

## 5. Experiential Account

The peer-reviewed literature on some aspects of OSIs is limited. One of the most neglected aspects is the lived experiences of those impacted by OSIs. As stated above, PTSD (as a prime OSI) results from exposure to a traumatic event(s). Intrinsically, military and police service involves traumatic experiences and in that, it is well understood and accepted that respective personnel may develop an OSI. Perhaps less understood are the impacts that these vocations' work environments have on the development of OSIs and in particular,



PTSD. Environmental work factors associated with a diagnosis of PTSD include challenges with faulty equipment, stressful interactions with colleagues, daily operational hurdles, and unclear roles and responsibilities [77]. Similarly, amongst a sample of British police officers, Collins and Gibbs found that these organizational factors—including increased workload, lack of support, and lack of communication and consultation—rather than police-specific duties were ranked as their highest stressors [78]. These findings suggest that a supportive work environment may offer a protective measure against developing PTSD, despite their exposures to critical events in the course of their duties [77]. Considering the impact that the work environment can have on military and police personnel, it cannot be overlooked and is an important topic to explore in relation to OSI treatment and related policy.

Just as a supportive work environment can act as a positive buffer against the impacts of traumatic events [79], so too can a supportive peer environment during OSI treatment and more generally for people with OSIs [80]. For military and police veterans, peer support and group therapy reinforce shared and common experiences and can help combat feelings of loneliness and feeling misunderstood [80]. Particularly during the transition from military and police service to civilian life, many veterans experience feelings of lost purpose and connection with peers [81] creating feelings of isolation, loneliness, and suicide [82]. During this period of transition, veterans often need to wrestle with losing a sense of their identity which they fused to their military membership during service [83]. Social engagement, learning new skills, and finding a sense of purpose can help offset these feelings [84]. Peer support can also help in introducing veterans to treatment programs, encourage their ongoing engagement, and enable the use of the skills acquired through therapy [66]. A caveat is that some military and police veterans prefer not to be involved with other veterans for various reasons or report facing challenges accessing peer support [66,85]. For some veterans, they avoid peer support due to a lack of trust for others, social anxiety, and feeling that the support is focused on social gatherings rather than on recovery [66].

OSI, and in particular PTSD, is typically known for its negative impacts on one's physical, mental, and social health and well-being. Yet an important but lesser studied impact of PTSD is post traumatic growth (PTG)—positive mental impacts experienced by the person following their exposure(s) to one or more traumatic event(s) [86]. PTG can include a positive re-alignment of priorities, finding more appreciation in daily experiences, and an improved ability to navigate difficulties [87]. Greater PTG has been associated with younger veterans, those with greater PTSD symptoms, and greater feelings of support [87]. These accounts of PTG, if shared, may enhance hope for members of the military and RCMP with PTSD that they can experience such positive personal growth and the need to find a new purpose following their traumatic service experiences. A notable example of personal growth following trauma is former Canadian Armed Forces Lieutenant-General, Roméo Dallaire (Dallaire). In 1994, Dallaire was the United Nations Commander during the Rwanda Genocide and when he returned home to Canada, experienced anger and an avoidance of social situations [88]. Dallaire attempted suicide and was eventually diagnosed with PTSD [89]. After 36 years of service in the military, his PTSD lead to his eventual discharge; a change that left him with feelings of abandonment and depression [88]. In 2004, Dallaire had the difficult task of testifying about the traumatizing crimes he witnessed in Rwanda, but once that task was behind him, he began focusing on a new sense of purpose and on projects to end the use of child soldiers around the world [88]. Since then, Dallaire has received numerous accolades including the Order of Canada and is an accomplished author [90]. While today, Dallaire still experiences triggering memories from notes, certain smells, and photographs [90], he uses medication and therapy to treat his PTSD symptoms [89]. Dallaire highlights that PTSD does not just have serious physical consequences, but can involve a moral injury that attacks the (figurative) heart and mind [91]. Dallaire attributes an atmosphere of love to realizing that his life was worth living, stating *"it is the most powerful absolute in humanity"* [90].

## 6. Conclusions

OSIs involve a range of mental disorders, often co-existing with other mental and physical health challenges [1,2]. At present, the main interventions used to treat OSIs are psychotherapies and psychotropic medications, as well as additional interventions such as neuromodulation, (neuro)cognitive remediation, yoga, and creative art therapies; social supports are also needed. Given the complex nature of OSIs and related comorbid conditions, a comprehensive and individualized care (treatment, rehabilitation, and more) plan is needed for best outcomes.

While OSIs encompass various mental disorders, PTSD is arguably the most well-known. While PTSD is present in the general Canadian population, with military and police personnel, it is more likely to develop as they often experience trauma in hostile and/or remote settings [24]. Although individuals with PTSD may experience ongoing mental symptoms throughout their lives, between 38 and 60% of these individuals show marked improvements with treatment [31–33] and some eventually experience PTG [86].

Although OSIs are common for Canadian military and RCMP personnel, the peer-reviewed literature concerning OSIs remains limited. More research and its dissemination is needed to further develop related best practices and enhance recovery for those diagnosed with this complex set of mental disorders and their comorbidities. This includes further study of the impacts that a supportive work environment may have on military and police personnel to try to prevent their development of PTSD, as well as of the impacts that peer support can have on their willingness to seek and collaborate with their care. Additionally, more research and its dissemination is needed concerning the lived experience of PTG, such as that of Dallaire; a greater awareness of this may enhance hope for military and RCMP personnel with PTSD and other OSIs so that they can experience positive personal growth following their impactful service injuries. Practice and policy related to people with OSIs are expected to benefit from such progress.

**Author Contributions:** All authors (A.R., A.S., S.L. and D.N.) contributed to the writing, review, and editing of this manuscript. All authors have read and agreed to the published version of the manuscript.

**Funding:** This research received no external funding.

**Institutional Review Board Statement:** Not applicable.

**Informed Consent Statement:** Not applicable.

**Conflicts of Interest:** The authors declare no conflict of interest.

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
