# Peer review of "Operational Stress Injury"

_encyclopedia, doi:10.3390/encyclopedia3040095_

Round 1

Reviewer 1 Report

Encylopeida-2616593

Thank you for the opportunity to review the manuscript titled: “Operational Stress Injury.” This entry covers the titular topic by providing information on the types of diagnoses and symptoms that commonly emerge from OSI’s. It goes on to discuss epidemiology and etiology, treatment interventions and protective factors, and the real life impact of these conditions.

The entry provides an appropriately broad and detailed level of coverage and is generally well-written. I have noted several relatively minor suggestions that I believe could help strengthen the entry before publication.

1.      In American military settings, there is a similar concept of “combat stress reaction” that is of course not as inclusive as the term “operational.” It might be worth addressing how the concept of OSI’s both allows for recognizing reactions beyond PTSD, and the multitude of operational stressors beyond combat itself that might impact both service members and law enforcement.

2.      Lines 35-36 note that: “[…]as long as the injury develops during, as a result of, or is worsened by, military/police service, then it can be considered an OSI.” Would this include injuries that are unrelated to service? For instance, grief from the loss of a loved one outside the service and absent any operational stressors, but during one’s active period of service? Or would it need to occur during actual operations? It may be worth clarifying either way.

3.      Line 87 uses the word “it” in a way that is ambiguous as to whether it refers to ASD or PTSD. It would help to replace “it” with PTSD or ASD.

4.      Line 87 makes the assertion that “it more commonly manifests with dissociative symptoms” without a corresponding citation. The manuscript would be strengthened by a citation.

5.      I think the manuscript would benefit from providing a bit more context to explain why anxiety disorders and substance-related disorders are common forms of OSI’s. The section Experiential Account gets into this a bit, but it could also help to connect some of these directly to the conditions noted earlier (besides PTSD, which is relatively self-evident).

6.      On a related note, I wondered about including some reference to the impact of the constant, unpredictable nature of threats in some operational roles. For instance, going to sleep expecting to be awoken at any moment by danger, or spending hours next to a radio waiting to get called out to an emergency with just moments to react. In my experience with service members, this has been a major factors (apart from criterion A trauma) mentioned as a source of persistent anxiety and hyperarousal. Being far from family and home, or local but with long, disruptive work hours, are some other experiences that come to mind that may contribute to OSI’s.

7.      Under psychotherapies for trauma-related nightmares, it may help to clarify that trauma-focused therapy also reduces nightmares (and a specific nightmare-focused treatment is generally not necessary).

8.      Cognitive behavioral conjoint therapy is a treatment for PTSD that is approached through couples attending sessions together, that may be worth noting in the section on family impacts.

9.      Trazodone for sleep and prazosin for nightmares appear (anecdotally) to be incredibly common medications for OSI’s. It may be worth noting those in the medication section.

10.  In section 4.3 regarding additional interventions, the article may be strengthened by briefly noting the evidence for/against attempts to re-integrate individuals with OSI’s into their work roles.

Reviewer 2 Report

Good summary. Can expand on link of Lt. Gen., ptsd, and need to find a purpose after exposure to stress. 

The manuscript answer positively all those issues. The only point was made in terms of linkages that is the comment to authors. Authors can choose to address this in a single sentence.

Author Response

Feedback for Reviewer # 2

Reviewer Comment: Good summary. Can expand on link of Lt. Gen., ptsd, and need to find a purpose after exposure to stress. 

The manuscript answer positively all those issues. The only point was made in terms of linkages that is the comment to authors. Authors can choose to address this in a single sentence.

Author Response: We would like to thank this reviewer for their helpful feedback on this manuscript. We have slightly reworded our section 5 of this document (experiential account) to account for the suggestion made above related to clarifying the need for those with OSIs to “find a purpose”.  

Reviewer 3 Report

Thank you for inviting me to review the entry Operational Stress Injury (OSI) for Encyclopedia. I found the draft entry appropriately detailed and informative. I think the authors captured the topic’s main idea and provided a balanced account of the diagnostic frameworks, epidemiology and etiology, and treatment. Below is specific feedback that should be considered in a revision.

Introduction

·      The term Operational Stress Injury is both non-clinical and non-medical. I would recommend the authors state this within the first paragraph.

·      OSI is generally applied to individuals with military or first responder backgrounds. The current draft fails to include other professions in the first responder category (e.g., firefighters, emergency medical technicians). Consider expanding the population beyond police officers.

·      Page 1, Line 42: What do the authors mean by “The study of OSIs is significant”? Are they suggesting that there is significance or importance in studying OSIs or that there has been significant study (i.e., research) on OSIs?

Diagnoses and Symptoms

·      Consider including the Diagnostic and Statistical Manual of Mental Disorders – 5 (DSV5) in the text. For example: “In this category, the three mental disorders that appear to be most commonly addressed and are in the DSM5 are…” (see page 2, line 65)

·      Page 2, Line 72: More recently, some researchers have suggested that one can acquire a fear response to stimuli via observational learning or verbal transmission. Consider looking more closely into this, as it closely relates to the population under consideration (i.e., military personnel and first responders).

·      Page 4, Lines 168-171: Consider a citation.

·      Page 4, Line 168: What is meant by “visible minorities”?

Biopsychosocial Care

·      Exposure-based approaches to treatment are frequently highlighted in the PTSD intervention literature (see with work of Edna Foa). Although often (but not always) a component of CBT, I think exposure-based interventions deserve their own paragraph in this section of the paper.

Round 2

Reviewer 1 Report

Thank you for the opportunity to review the revised version of the manuscript. The authors have satisfactorily addressed my comments from the previous version.

Reviewer 2 Report

no comments

Reviewer 3 Report

Thank you for inviting me to review the revision of this entry. I believe the authors have done an outstanding job addressing the feedback. The entry is more complete.